# AMPNet: Attention as Message Passing for Graph Neural Networks

## Abstract

Graph Neural Networks (GNNs) have emerged as a powerful representation learning framework for graph-structured data. A key limitation of conventional GNNs is their representation of each node with a singular feature vector, potentially overlooking intricate details about individual node features. Here, we propose an Attention-based Message-Passing layer for GNNs (AMPNet) that encodes individual features per node and models feature-level interactions through cross-node attention during message-passing steps. We demonstrate the abilities of AMPNet through extensive benchmarking on real-world biological systems such as fMRI brain activity recordings and spatial genomic data, improving over existing baselines by 20% on fMRI signal reconstruction, and further improving another 8% with positional embedding added. Finally, we validate the ability of AMPNet to uncover meaningful feature-level interactions through case studies on biological systems. We anticipate that our architecture will be highly applicable to graph-structured data where node entities encompass rich feature-level information.

## 1 Introduction

Recent advancements in Deep Learning (DL) have fueled an explosion of successful applications on a broad range of tasks where data is represented in Euclidean spaces. GNNs (Gori et al., 2005; Scarselli et al., 2008) have extended this success to non-Euclidean, graph-structured data, with applications in domains such as social networks (Cho et al., 2011), molecular graphs (You et al., 2018), biological networks (Zitnik et al., 2018), and traffic forecasting (Bai et al., 2021). GNNs operate on a *message passing* principle, allowing for nodes to pass information to neighboring nodes which can then be used to update hidden states attributed to either nodes or edges. This allows GNNs to be applied to multiple tasks, including node classification, edge prediction, and graph classification.

Many data domains exist where node entities exhibit rich or high-dimensional representations that can be broken down (e.g. genes inside of cells, words inside of documents, patches inside of an image), and interactions between node features play a nontrivial role in graph learning tasks . Many GNNs are unable to capture interactions between individual features across different nodes, which may limit their expressiveness in tasks where feature interactions play an important role between different node entities. Other works have explored either individual feature importance or node-level interactions in various ways: GNNExplainer (Ying et al., 2019) proposed to find subgraph explanations of input graphs by framing explanation as a mutual information maximization problem. Graph Attention Networks (GATs) (Veličković et al., 2017) introduced interpretability directly into the model through self-attention (Bahdanau et al., 2014; Luong et al., 2015), computing edge-level attention coefficients between nodes during message-passing. These methods, however, fall short of modeling inter-feature interactions across different nodes. Furthermore, in scenarios where additional information is available for individual node features (i.e. positional encoding or word embeddings), options for encoding the additional information often rely on concatenation and additional encoding modules, which adds complexity to the architecture and still bottlenecks feature representation into a single embedding vector per node.

To address this, we propose an interpretable message-passing framework that can uncover feature-level interactions across neighboring nodes. Instead of computing attention between node embeddings, we embed individual features for each node and compute attention between *feature* embeddings for adjacent nodes, visualized in Figure 1. To compute inter-feature attention, a multi-head

attention mechanism (Vaswani et al., 2017) is applied on each edge during message passing using the source node feature vector set as the context sequence and the destination node feature set as the target sequence, yielding interpretable attention coefficients between features of adjacent nodes. We call this approach *attention as message-passing*, and our proposed architecture which employs it AMPNet. We formulate AMPNet as a flexible GNN layer, able to integrate with other GNN layers and objectives while providing meaningful feature-level attention. Our approach is inherently interpretable, and allows for additional positional or learned feature embeddings to be embedded per individual node feature without bottlenecking node representation to a single vector.

We evaluate AMPNet against strong baseline models on several public benchmark and real-world datasets, including fMRI brain activity recordings, spatial transcriptomics, citation networks, and a newly constructed image network. AMPNet provides inter-feature attention for each edge of an input graph, which we inspect after training in interpretability case studies (Section 5). We show that in fMRI recordings, the obtained attention highlights differences in functional region attention between patients of varying age and health condition, and in mouse hippocampal tissue the attention highlights gene interactions between cells, suggesting potential biological interactions based on the identity of the neighboring cells.

## 2 RELATED WORKS

Previous approaches for introducing interpretability in GNNs can be divided into two main categories: (i) explainability methods which aim to explain the predictions of a GNN through post-hoc analysis, and (ii) interpretable models which yield explanations that are human-interpretable.

### 2.1 POST-HOC EXPLAINABILITY METHODS

Initial attempts to obtain post-hoc explanations for GNNs (Pope et al., 2019) adapted prominent explainability techniques from Convolutional Neural Networks (CNNs) to graphs, including gradient-based saliency maps (Simonyan et al., 2013), class activation maps (Zhang et al., 2018) among others (Zhang et al., 2018; Sundararajan et al., 2017; Selvaraju et al., 2017). These methods analyzed gradients or activation values in feature maps to determine the importance of model inputs.

Perturbation-based methods examine changes in model predictions under various input perturbations to measure feature importance (Luo et al., 2020; Yuan et al., 2021). Closest to our work is GNNExplainer (Ying et al., 2019), which learns a subgraph by maximizing mutual information to the original graph and provides a node feature importance mask. Other works adopt a generative approach for creating explanations, using deep neural networks (Luo et al., 2020) or generative flow networks (Li et al., 2023) to parameterize the generation process.

Surrogate methods aim to approximate the outputs of a complex black-box model with a less complex, interpretable model which could then be analyzed (Huang et al., 2022; Vu & Thai, 2020). Applying surrogate models to graph data is challenging due to the discrete nature of node features and topological information contained in the graph structure.

### 2.2 INTERPRETABLE MODELING

Interpretable modeling aims to design models that directly give explanations of their predictions, such as decision trees and attention-based models (Bahdanau et al., 2014; Gehring et al., 2016). Recent works caution against calling attention weights a completely faithful interpretation of token importance (Jain & Wallace, 2019; Wiegreffe & Pinter, 2019). While attention weights do not provide completely faithful interpretations, we do expect well-trained attention weights to uncover meaningful relationships among tokens (Wiegreffe & Pinter, 2019).

GATs (Veličković et al., 2017) introduced attention to GNNs by computing a self-attention among the neighborhood of a given node, yielding edge-level attention scores which then are used during message passing. We note that this attention computation operates on node feature vectors rather than among individual features, thus falling short of inferring feature-level relationships across nodes. Other works built on this by adapting GATs to heterogeneous graphs (Wang et al., 2019), sparsifying the graph attention mechanism (Zheng et al., 2020), and using spiking neural networks to provide another method for inexpensive graph attention computation (Wang & Jiang, 2022).

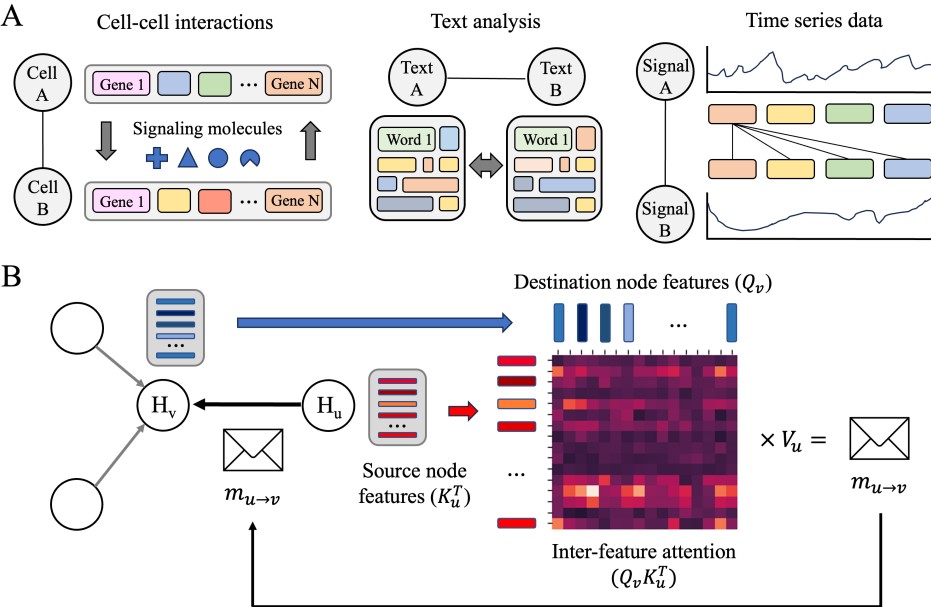

Figure 1: Framework overview. (A) Examples of data domains where nodes encode rich information about different entities, with additional information to encode per features. (B) Overview of proposed attention as message passing framework. AMPNet computes attention between feature sets of adjacent nodes, providing interpretable coefficients for feature-level interactions between nodes.

## 3 PROPOSED METHOD: ATTENTION AS MESSAGE PASSING

We begin by introducing our notation scheme for message-passing GNNs. Let $\mathcal{G} = (\mathcal{V}, \mathcal{E})$ denote a graph with node features $x_v \in \mathbb{R}^F$ for each node $v \in \mathcal{V}$, where $F$ is the number of features per node. GNNs iteratively pass messages between neighboring nodes connected by the set of edges $\mathcal{E}$, and in the process use both node features and graph structure to learn node representations $h_v \in \mathbb{R}^D$, where D is the hidden dimension of node embeddings. After $k$ message-passing iterations, node representation $h_v$ will contain information from its $k$-hop neighborhood within the graph. The general update rule for the $k$-th layer of a GNN can be represented as follows:

$$h_{\mathcal{N}(v)}^{(k)} = \texttt{AGGREGATE}^{(k)}\big(\big\{h_u^{(k-1)}, u \in \mathcal{N}(v)\big\}\big) \tag{1}$$

$$h_v^{(k)} = \texttt{COMBINE}^{(k)}\big(h_{\mathcal{N}(v)}^{(k)}, h_v^{(k-1)}\big), \tag{2}$$

where $\mathcal{N}(v)$ denotes the neighborhood of node $v$ and $h_v^{(k)}$ is the representation of node $v$ in layer $k$. The choice of AGGREGATE and COMBINE vary among different GNN architectures, with the constraint that AGGREGATE should be a permutation-invariant aggregator. A readout function is used to map learned node representations into predictions for feature, node, or graph-level tasks.

### 3.1 FEATURE EMBEDDING

Given input node features $x_v \in \mathbb{R}^F$, we define a mapping $\tau : \mathbb{R}^F \to \mathbb{R}^{F \times D}$ to transform node feature values into a set of feature vectors $\tau(x_v) = H_v \in \mathbb{R}^{F \times D}$ which will represent the node during the attention message passing step. Note that the embedding process may be task-dependent; for example, time-varying node features may have positional embedding added as part of the embedding process, or language data might have word vectors initialize the feature embedding.

In practice, we utilize positional encoding for timeseries and image datasets, and use a learned embedding table for feature representation in experiments on spatial genomics datasets and citation

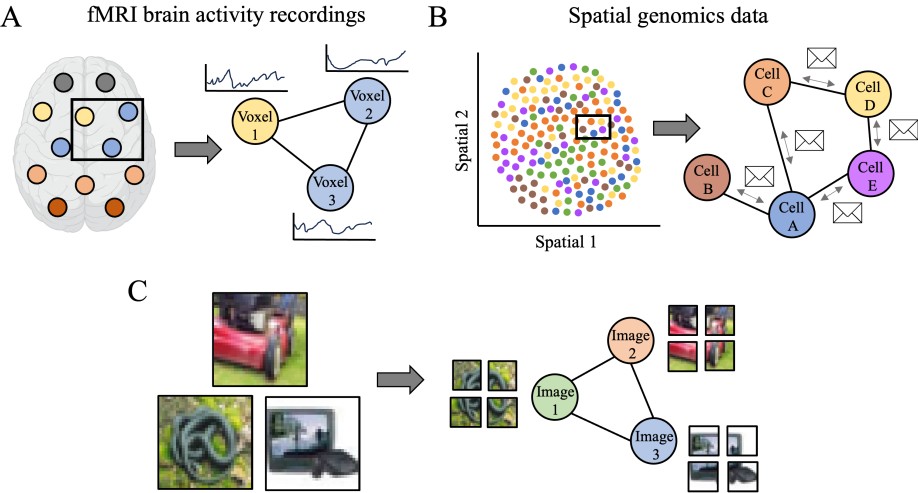

Figure 2: Visualization of graph creation process. (A) For fMRI recordings, the brain is parcellated into 424 regions, which are connected using a K-nearest neighbors graph according to the 3D coordinates of the centroid of each brain region. (B) For spatial genomics data, cells are connected to their nearest neighbors according to 2D coordinates from the original tissue capture. (C) For images, a network is created by probabilistically creating edges between images, with same-class images more likely to be connected.

networks. We concatenate the feature information encoding (i.e. positional embedding or learned embedding vector) with a projection of the feature value to obtain the final embedding representing that feature. For the case of positional embedding, this can be represented as:

$$H_v = \tau(x_v) = \text{CONCATENATE}[PE, x_v W_p], \tag{3}$$

where $PE \in \mathbb{R}^{F \times D}$ represents the positional embedding for the sequence of features for node $v$, $x_v \in \mathbb{R}^{F \times 1}$ is the input node features, and $W_p \in \mathbb{R}^{1 \times D}$ represents the projection matrix for feature values. We use sum concatenation to combine feature information and value projection.

### 3.2 ATTENTION-BASED MESSAGE CREATION

At a given message passing iteration $k$, we employ multi-head attention (Vaswani et al., 2017) as the message function between source node $u$ and destination node $v$. We give source node $u$'s feature vectors $H_u$ as the context sequence, and destination node $v$'s feature vectors $H_v$ as the target sequence. We can formulate the cross-node attention and update step as:

$$Q = H_v^{(k-1)} W_q; K = H_u^{(k-1)} W_k; V = H_u^{(k-1)} W_v \tag{4}$$

$$H_{\mathcal{N}(v)}^{(k)} = \underset{u \in \mathcal{N}(v)}{\text{AGGREGATE}} \left( \text{softmax} \left( \frac{QK^\top}{\sqrt{D}} \right) V \right) \tag{5}$$

$$H_v^{(k)} = \text{CONCATENATE}[H_v^{(k-1)}, H_{\mathcal{N}(v)}^{(k)}] W_c + b_c, \tag{6}$$

where $W_q, W_k, W_v$ are weight matrices for query, key, and value vectors, $H_{\mathcal{N}(v)}^{(k)}$ represents the set of feature vectors obtained from aggregating node $v$'s local neighborhood, and $W_c$ and $b_c$ represent the weights and biases for the update function. We note that a deeper multi-layer transformer decoder may be used rather than a simple multi-head attention layer to increase expressivity in the message-passing step. Additionally, in practice we compute cross-attention on edges in the graph in batch-wise fashion rather than all at once during the message-passing step to control the computational footprint of message passing. The full AMPNet algorithm is summarized in Algorithm 1.

---

**Algorithm 1** AMPNet message-passing algorithm

---

**Require:** Graph $\mathcal{G} = (\mathcal{V}, \mathcal{E})$, input features $h_v^0 \in \mathbb{R}^F$, feature embedding table $W_E$
1: $H_v^0 \leftarrow \text{CONCATENATE}[W_E \parallel h_v]$
2: **for** node $v \in N$ **do**
3:      **for** node $u \in \mathcal{N}(v)$ **do**
4:          $Q = H_v^{(k-1)} W_q$
5:          $K = H_u^{(k-1)} W_k$
6:          $V = H_u^{(k-1)} W_v$
7:      **end for**
8:      $H_{\mathcal{N}(v)}^{(k)} = \underset{u \in \mathcal{N}(v)}{\text{AGGREGATE}} \left( \text{softmax} \left( \frac{QK^\top}{\sqrt{D}} \right) V \right)$
9:      $H_v^{(k)} = \text{CONCATENATE}[H_v^{(k-1)}, H_{\mathcal{N}(v)}^{(k)}] W_c + b_c$
10: **end for**

---

## 4 EXPERIMENTS

We evaluate AMPNet in self-supervised reconstruction settings on three real-world biological datasets, as well as on a newly constructed image network dataset. We also evaluate AMPNet in a link prediction task on a standard citation network dataset. More details about datasets and hyperparameters used are available in Appendix A.1 and A.5.

### 4.1 DATASETS

We first evaluate the performance of AMPNet in self-supervised reconstruction of fMRI recordings from the publicly-available UK Biobank (UKB) (Miller et al., 2016) dataset. The UKB provides task-based and resting-state functional magnetic resonance imaging (fMRI) recordings from subjects aged 40-69 years old. After standard preprocessing, we take brain activity recordings and divide them into 424 brain regions using the AAL-424 atlas (Nemati et al., 2020).

Next, we benchmark AMPNet on a masked gene expression prediction task on two spatial transcriptomics datasets. The Slideseq-V2 spatial transcriptomics dataset (Stickels et al., 2021) is a mouse hippocampal dataset consisting of $41,786$ cells and $4,000$ genes, with each cell being categorized into one of 14 different cell types. We also benchmark on a 10X Genomics spatial transcriptomics dataset consisting of 4247 cells from human heart tissue expressed in 36601 genes. For both datasets, we follow conventional preprocessing and normalization procedures for spatial genomics data.

To evaluate AMPNet on edge-level prediction, we use the OGBN-arXiv citation network dataset (Wang et al., 2020), which comprises of $169,343$ nodes representing computer science papers in the Microsoft Academic Graph (Wang et al., 2020) and $116,6243$ edges. We reprocess the OGBN dataset in order to obtain word feature identities for attention analysis in A.4.

Finally, we evaluate AMPNet's ability to do masked image reconstruction on a new image network dataset, which we construct from the Cifar-100 image dataset (Krizhevsky et al., 2009). In the image network, each node represents an entire original image in the dataset, connected probabilistically to other images of the same or different class. This is a particularly challenging graph learning problem given the high-dimensional features of an image in pixel space as well as the semantic information present in images which are relevant for image-based tasks. AMPNet is well-suited for this task given its ability to encode multiple image patches tokens per node along with 2D positional encoding per patch token. More details about image network creation can be found in Appendix A.1.

### 4.2 BASELINE METHODS

We compare AMPNet against a set of popular message-passing architectures (GCN (Kipf & Welling, 2016), GraphSAGE (Hamilton et al., 2017), GAT(Veličković et al., 2017)) on self-supervised reconstruction tasks. We additionally compare against GraphMAE (Hou et al., 2022), a recent graph autoencoder technique which uses masked learning objectives, as well as GPS Graph Transformer, a SOTA graph transformer (Rampášek et al., 2022).

Table 1: Benchmark on UK Biobank fMRI recording reconstruction. Performance is reported across 5 runs in terms of Mean Squared Error (MSE) and $R^2$. AMPNet improves upon baselines by $20\%$ with no positional encoding, and by $28\%$ with positional encoding added.

| Method | Masking Strategy | MSE ($\downarrow$) | $R^2$ ($\uparrow$) |
|---|---|---|---|
| GCN | Replace noise | $0.789 \pm 0.00016$ | $0.190 \pm 0.00017$ |
| | Fill in mean | $0.748 \pm 0.00026$ | $0.232 \pm 0.00026$ |
| | Linear interpolation | $0.779 \pm 0.00047$ | $0.200 \pm 0.00049$ |
| GraphSAGE | Replace noise | $0.754 \pm 0.00160$ | $0.226 \pm 0.00164$ |
| | Fill in mean | $0.686 \pm 0.00108$ | $0.296 \pm 0.00111$ |
| | Linear interpolation | $0.736 \pm 0.00161$ | $0.244 \pm 0.00166$ |
| GAT | Replace noise | $0.781 \pm 0.00035$ | $0.198 \pm 0.00036$ |
| | Fill in mean | $0.741 \pm 0.00020$ | $0.239 \pm 0.00020$ |
| | Linear interpolation | $0.771 \pm 0.00016$ | $0.208 \pm 0.00016$ |
| GraphMAE | Replace noise | $0.813 \pm 0.00053$ | $0.165 \pm 0.00054$ |
| | Fill in mean | $0.782 \pm 0.00028$ | $0.197 \pm 0.00029$ |
| | Linear interpolation | $0.809 \pm 0.00036$ | $0.170 \pm 0.00037$ |
| GPS Graph Transformer | Replace noise | $0.801 \pm 0.00276$ | $0.178 \pm 0.00283$ |
| | Fill in mean | $0.767 \pm 0.00276$ | $0.212 \pm 0.00284$ |
| | Linear interpolation | $0.810 \pm 0.00725$ | $0.170 \pm 0.00743$ |
| AMPNet | Tokenization | $0.485 \pm 0.00026$ | $0.501 \pm 0.00027$ |
| | Tokenization + PE | $\mathbf{0.410 \pm 0.00106}$ | $\mathbf{0.578 \pm 0.00109}$ |

## 4.3 Experimental Setup

We formulate our self-supervised reconstruction tasks to provide a fair comparison of AMPNet against baseline methods to verify the advantages of AMPNet's feature embedding procedure. For fMRI recording reconstruction, we construct a KNN graph using the 3D coordinates of each brain region. We tokenize the recording into patches of 20 timepoints and train AMPNet to reconstruct 50% of the tokens per voxel. For baseline architectures, we provide the 400 timepoint series as an input vector with masked patches filled in by (i) replacing with random noise, (ii) replacing with the mean value, and (iii) interpolating between adjacent unmasked points.

For masked gene expression tasks, we learn an embedding table with a unique vector for each gene in the dataset. We sample a fixed number of nonzero genes per cell with replacement, and embed genes using its corresponding learned embedding along with its expression value projection. In practice, we sample 50 and 30 nonzero genes with replacement for the mouse hippocampus (Stickels et al., 2021) and human heart datasets respectively, and use a masking ratio of 20 percent for both datasets.

On the masked image reconstruction task, we follow the same masking procedure and positional encoding scheme as the Masked Autoencoder (He et al., 2022). We use a 2-layer AMPNet encoder and an additional AMPNet layer as the decoder for the framework, and follow the convention of applying the encoder to only unmasked patches. For each image we use a patch size of 4x4 pixels with a masking rate of 20% on each image.

## 4.4 Results

The results of our benchmarking experiments on fMRI recording reconstruction are summarized in Table 1. AMPNet improves over all baselines in reconstructing fMRI signals, including the recent GraphMAE (Hou et al., 2022) masked graph autoencoder approach and a strong graph transformer baseline (Rampášek et al., 2022). We note that with the addition of positional encoding, AMPNet outperforms the closest baseline method by 28% in terms of $R^2$. When we remove the positional encoding, performance gains decrease but are still ahead of baselines by 20%. This is strong evidence that feature embedding gives AMPNet strong reconstruction ability and that performance is further enhanced by augmenting timepoint patches with relative positional embedding.

Table 2: Experimental results on masked gene expression prediction on the human heart and mouse hippocampus spatial genomics datasets. Performance is reported across 5 runs in terms of MSE and $R^2$. AMPNet outperforms baseline methods on predicting masked gene expression values on both datasets.

| Method | Mouse Hippocampus | | Human Heart | |
|---|---|---|---|---|
| | MSE ($\downarrow$) | $R^2(\uparrow)$ | MSE ($\downarrow$) | $R^2(\uparrow)$ |
| GCN | $0.0178 \pm 0.00056$ | $0.264 \pm 0.00930$ | $0.0015 \pm 0.00005$ | $0.776 \pm 0.01481$ |
| GraphSAGE | $0.0144 \pm 0.00038$ | $0.387 \pm 0.02043$ | $0.0016 \pm 0.00012$ | $0.778 \pm 0.01537$ |
| GAT | $0.0185 \pm 0.00120$ | $0.237 \pm 0.02509$ | $0.0015 \pm 0.00006$ | $0.770 \pm 0.01759$ |
| GraphMAE | $0.0178 \pm 0.00044$ | $0.271 \pm 0.02661$ | $0.0015 \pm 0.00014$ | $0.762 \pm 0.00568$ |
| AMPNet | $\mathbf{0.0096 \pm 0.00077}$ | $\mathbf{0.562 \pm 0.03197}$ | $\mathbf{0.0011 \pm 0.00005}$ | $\mathbf{0.832 \pm 0.01032}$ |

Table 3: Experimental results for link prediction on OGBN-arXiv and masked image reconstruction on Cifar-100. Performance is measured in terms of Area Under ROC curve (AUROC) for link prediction, and in terms of MSE and R2 for masked image reconstruction. GraphMAE is not benchmarked on link prediction since it is designed for node and graph-level classification.

| Method | OGBN-arXiv | Cifar-100 | |
|---|---|---|---|
| | AUROC ($\uparrow$) | MSE ($\downarrow$) | $R^2(\uparrow)$ |
| GCN | $85.6 \pm 0.04$ | $0.673 \pm 0.003$ | $0.333 \pm 0.006$ |
| GraphSAGE | $86.1 \pm 0.07$ | $0.473 \pm 0.003$ | $0.526 \pm 0.008$ |
| GAT | $85.4 \pm 0.10$ | $0.668 \pm 0.016$ | $0.331 \pm 0.018$ |
| GraphMAE | - | $0.753 \pm 0.010$ | $0.247 \pm 0.002$ |
| AMPNet | $\mathbf{89.5 \pm 0.13}$ | $\mathbf{0.306 \pm 0.010}$ | $\mathbf{0.696 \pm 0.008}$ |

On experiments on spatial transcriptomics datasets, AMPNet again outperformed baseline methods at reconstructing gene expression values on both datasets (Table 2). We hypothesize that the learned gene embedding involved in the feature embedding procedure for AMPNet allows it to learn a representation space for different genes which encodes them more efficiently and gives AMPNet the potential to learn interactions between genes. We note that by sampling nonzero genes, AMPNet avoids modeling zero-expressed genes which are not present in cells, and has the potential to represent nodes with a variable number of genes.

We evaluate AMPNet's performance on a self-supervised link prediction in Table 3. On the edge-level task, AMPNet effectively learns representations which allow it to improve on other baseline methods by 3.4% in terms of AUROC. Note that we do not evaluate GraphMAE's performance on link prediction since it was originally designed and benchmarks for node and graph-level classification tasks (Hou et al., 2022).

For masked image reconstruction, we evaluate AMPNet's ability to effectively tokenize entire images per node and perform reconstruction of masked patches in Table 3. AMPNet performs 17% better than the closest baseline method in terms of $R^2$ at reconstructing masked pixel values in Cifar-100 images. We hypothesize that the tokenization approach, which closely follows standard tokenization procedures in Computer Vision, allows AMPNet to encode images more effectively in patches. Additionally, 2D positional embedding gives AMPNet more information about the relative position of different image patches compared to conventional GNN approaches. We believe that this is a promising approach for learning on larger-resolution image networks which may exist in social media networks or other domains.

## 5 INTERPRETABILITY

We analyze the inter-feature attention obtained from AMPNet during message passing by designing case studies which aim to answer the following question: does feature-level attention uncover meaningful node feature interactions across different datasets and tasks?

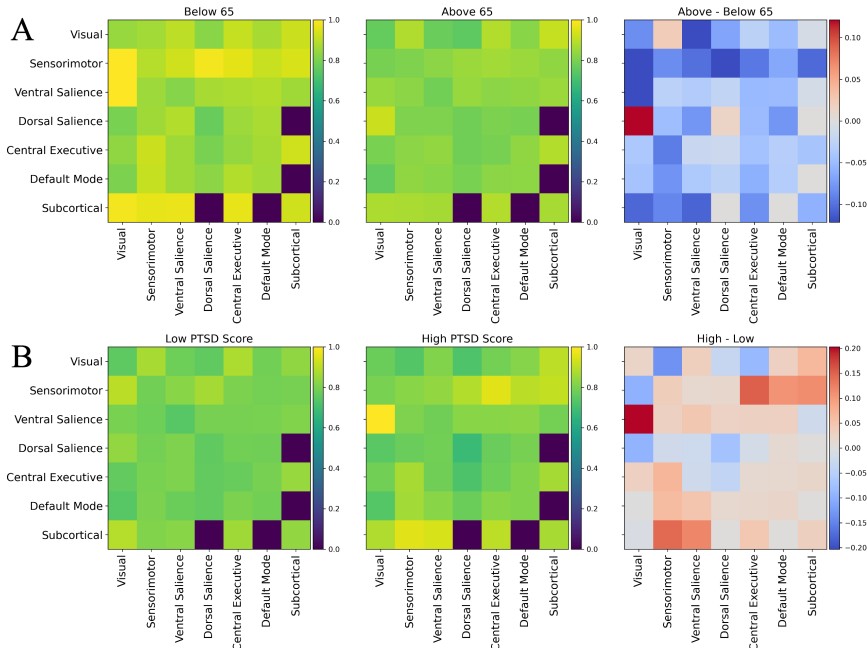

Figure 3: Visualizations of AMPNet feature-level attention between different functional groups in the brain. (A) Averaged attention heatmaps between functional regions of the brain for different age populations, with the difference in attention by age group visualized on the right subplot. (B) Similar heatmaps visualized for post-traumatic stress disorder (PTSD) scores, highlighting differences in attention in patients with low vs high PTSD score.

## 5.1 CASE 1: FUNCTIONAL REGION ATTENTION IN FMRI RECORDINGS

During message passing on the fMRI recording-constructed graph, AMPNet generates cross-attention matrices between brain voxels and their 5 closest neighbors in the K-neighbor graph. We group the 424 brain voxels into 7 functional regions, namely the visual, sensorimotor, ventral salience, dorsal salience, central executive, default mode, and subcortical regions of the brain. Taking 100 unseen test set recordings, we extract attention matrices between all connected nodes, average the attention matrices across timepoints per node, and split patient recordings according to conditions such as Age and post-traumatic stress disorder (PTSD) score. We then average attention values across patient recordings with the same condition, and aggregate the node attention into the 7 functional regions, allowing us to examine differences in functional region attention between patients with different conditions.

In Figure 3A, the attention between functional regions is shown between patients below 65 years of age (left) and those above 65 (middle). The difference in attention between the two groups, as visualized on the rightmost plot, indicates that older patients tend to have higher attention between the dorsal salience regions and visual cortex regions. This follows previous literature that shows changes in dorsal pathways as people age (Yan et al., 2023). Furthermore, Figure 3B shows similar visualizations for patients with high and low PTSD scores, revealing higher attention between sensorimotor areas and central executive, and subcortical areas. This also follows previous literature on the somatosensory basis of PTSD, where arousal and higher-order capacities get affected (Kearney & Lanius, 2022). These patterns in attention reveal potential differences in functional region attention picked up by AMPNet among patients of varying conditions during unsupervised training on fMRI recordings.

## 5.2 CASE 2: GENE INTERACTIONS IN SPATIAL TRANSCRIPTOMICS

In spatial genomics datasets, each node corresponds to a cell which is represented by a set of expressed genes. During message-passing AMPNet provides attention matrices representing interac-

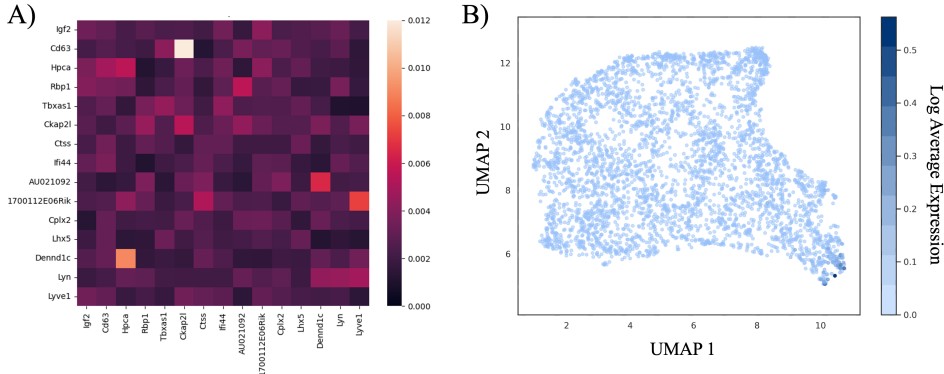

Figure 4: (A) Averaged attention between 15 genes across edges connecting neighboring astrocyte cells in the mouse hippocampus dataset. (B) UMAP of learned gene embeddings from AMPNet, colored by average expression value of each gene across astrocyte cells.

tions between genes of different cells. Gene interactions receiving higher attention between nodes can highlight possible biological connections which can be avenues of potential further exploration in the data. For example, Figure 4A shows an averaged attention heatmap across all self-edges connecting astrocyte cells in a subgraph sampled from the mouse hippocampus dataset (Stickels et al., 2021). This astrocyte-astrocyte feature-level attention matrix identifies a key interaction between CD63, a member of the tetraspanin family of cell surface proteins, and CKAP2L, a mitotic spindle protein controlling cellular division. Previous work has identified that CD63 may be either pro- or anti-tumorigenic, depending on tissue context (Dey et al., 2023). CD63 expression is also highly enriched in glioblastoma, a highly lethal malignancy of the astrocytes, and may play a role in progression of these cancers (Aaberg-Jessen et al., 2018). Our data hint that CD63 may play an important role in controlling cellular division through astrocyte-astrocyte cellular communication, which may represent an exciting new target for antitumoral agents.

Figure 4B shows a UMAP embedding of the gene embeddings learned by AMPNet in an unsupervised manner during training. Each vector in the embedding table represents one gene in AMPNet's vocabulary of known genes, and genes with similar function or importance are expected to be closer together in the learned embedding space. Each gene in Figure 4B is colored by its average expression value across all astrocyte cells in the mouse hippocampus dataset. We see that the learned embeddings form distinct structures during training, and that highly-expressed genes for astrocytes are clustered together. We hypothesize that this ability to learn gene vectors in embedding space and contextualize them for different cell types allows AMPNet to outperform other methods in gene expression prediction tasks.

## 6 CONCLUSION

In this work, we propose AMPNet, a novel message passing framework for GNNs that is able to uncover meaningful feature-level interactions between neighboring nodes. We demonstrate the utility of AMPNet's feature embedding approach for encoding rich information about node features across different data domains. In benchmarking experiments, we show that AMPNet outperforms all other methods in self-supervised tasks, and that the performance gains are more significant when positional encoding or other embeddings is added. We analyze the feature-level interactions uncovered by our model on real-world biological datasets in two case studies.

There are several avenues for improvement upon the AMPNet operator, which can be addressed in future work. A more efficient selection strategy for node features in sparse datasets such as those seen in spatial genomics might yield better node representations during training compared to simple nonzero feature sampling. Additionally, sparse or linear attention mechanisms can be implemented to improve the efficiency and overhead of the architecture. Finally, incorporating edge features, or possibly features assigned to the relationship between specific node features, may also be an interesting direction for further contextualizing feature-level interactions in graph-structured data.

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
