# A APPENDIX

## A.1 DATASET DETAILS

We first evaluate the performance of AMPNet in self-supervised reconstruction of fMRI recordings from the publicly-available UK Biobank (UKB) (Miller et al., 2016) dataset. The UKB provides task-based and resting-state functional magnetic resonance imaging (fMRI) recordings along with medical records from over 40,000 subjects aged 40-69 years old. We take 1000 brain activity recordings from UKB and use a 80/10/10 split for training, validation, and test respectively. Standard preprocessing was done on all recordings, including motion correction, normalization, temporal filtering, and ICA denoising (Salimi-Khorshidi et al., 2014; Abdallah, 2021). We used a parcellation of the brain into 424 brain regions using the AAL-424 atlas (Nemati et al., 2020), and normalized the recordings per voxel by subtracting the mean of each voxel and scaling by the standard deviation of signal values from all 1000 recordings.

Next, we evaluate AMPNet on a masked gene expression prediction task on two spatial transcriptomics datasets. The Slideseq-V2 spatial transcriptomics dataset (Stickels et al., 2021) is a mouse hippocampal dataset with high RNA capture efficiency and near-cellular spatial resolution. The data consists of $41,786$ cells, expressed in $4,000$ genes, with each cell being categorized into one of $14$ different cell types. We obtain the dataset from the Squidpy library (Palla et al., 2022), and follow standard preprocessing and normalization procedures, including normalizing expression values roughly between 0 and 1 by scaling with the 99th percentile of the data. We also benchmark on a 10X Genomics spatial transcriptomics dataset consisting of human heart tissue following the same normalization procedure.

Finally, we also evaluate AMPNet's ability to do masked image reconstruction on an image network dataset constructed from the Cifar-100 image dataset, which contains 60k images belonging to 100 classes (Krizhevsky et al., 2009). We split the Cifar-100 dataset into 40k/10k/10k train/val/test image splits, and connect training set images to each other with $2\%$ probability for same-class images, and $0.01\%$ probability for images of different classes. This yields a training graph with an average node degree of 12, with most image neighbors belonging to the same class. We create separate image graphs for the validation and test sets, and adjust probabilities to maintain the average node degree of 12.

We additionally train AMPNet on a link prediction task on the OGBN-arXiv dataset (Wang et al., 2020), which comprises of $169,343$ nodes representing computer science papers in the Microsoft Academic Graph (Wang et al., 2020) and $116,6243$ edges. In order to obtain the identity of word features for attention analysis, we preprocess a new version of the dataset by applying lemmatization and stop word removal to the raw abstract text of the OGBN-arXiv documents. We then selected the top $100$ words in the dataset according to which words had the highest variance in TF-IDF score across all nodes in the dataset. This yielded a word count matrix in the form of bag-of-words node features, where word identities were available for inspection in attention analysis. The learned embeddings and attention weights between nodes are inspected in interpretability case study **??**.

We benchmark AMPNet on several standard benchmark datasets for unsupervised node classification. The Cora dataset (Sen et al., 2008) consists of $2,708$ nodes each comprising a scientific paper, with $5,429$ edges representing citation links, 7 classes, and $1,433$ features per node. Pubmed (Sen et al., 2008) similarly contains $19,717$ nodes, $44,338$ edges, 3 classes, and 500 node features. Amazon Computers (Shchur et al., 2018) is a segment of the Amazon co-purchase graph, with $13,381$ nodes representing goods and edges denoting items frequently bought together.

## A.2 EXPERIMENTS SETUP DETAILS

For fMRI recording reconstruction, we construct a graph of brain regions, by connecting each region to its 5 nearest neighbors based on the xyz coordinates. We implement feature embedding for AMPNet by tokenizing 400 timepoints of signal per brain region into 20 patches of 20 timepoints, with positional encoding optionally added. We mask 50% of the tokens per voxel and train AMPNet to reconstruct the missing signal tokens. For baseline architectures which represent nodes as a single vector, we provide the 400 timepoint series as an input vector, and benchmark against three options for masking timepoint patches: (i) replacing masked signal patches with random noise, (ii)

replacing values with the mean signal value, and (iii) interpolating between adjacent visible signal values around the masked timepoint patch.

For masked gene expression tasks, many genes are zero expressed and do not convey information about the state of a cell. We therefore sample a fixed number of nonzero expressed genes per cell with replacement, and encode these to represent a cell in AMPNet. For baseline architectures, the data is given as a vector input for each cell, with the length of the vector corresponding to the total number of genes in the dataset. Note that AMPNet's representation of the expression data is more flexible, since the number of genes in the dataset does not determine the input size of data in AMPNet. In practice, we sample 50 and 30 nonzero genes with replacement for the mouse hippocampus (Stickels et al., 2021) and human heart datasets respectively, and use a masking ratio of 20 percent for both datasets.

Table 4: Experimental results on node classification. For self-supervised methods, models are pre-trained using a feature reconstruction objective, and test set accuracy of a linear classifier is reported based on learned node representations. The available data for each training method is shown in the second column, where **X** and **A** denote node features and graph adjacency information, respectively.

| Method | Available Data | Cora | Pubmed | Amazon Computers |
|---|---|---|---|---|
| Raw Features | **X** | $47.3 \pm 2.27$ | $63.7 \pm 3.13$ | $85.7 \pm 1.12$ |
| DeepWalk | **A** | $70.1 \pm 1.42$ | $64.5 \pm 1.97$ | $86.3 \pm 0.95$ |
| Random-init | **X, A** | $70.2 \pm 1.20$ | $66.3 \pm 3.83$ | $84.7 \pm 0.79$ |
| DGI | **X, A** | $69.6 \pm 2.36$ | $68.2 \pm 4.14$ | $83.9 \pm 1.34$ |
| GRACE | **X, A** | $\mathbf{80.3 \pm 1.99}$ | $76.0 \pm 4.10$ | $81.4 \pm 0.63$ |
| BGRL | **X, A** | $79.7 \pm 1.69$ | $73.4 \pm 3.71$ | $\mathbf{84.9 \pm 1.03}$ |
| GraphMAE | **X, A** | $78.7 \pm 2.02$ | $73.8 \pm 1.86$ | $74.9 \pm 1.75$ |
| AMPNet (masking) | **X, A** | $73.0 \pm 3.10$ | $\mathbf{76.2 \pm 0.88}$ | $77.8 \pm 2.07$ |

## A.3 SUPPLEMENTARY RESULTS

We additionally provide benchmarks on standard datasets for unsupervised node classification against recent contrastive graph-based architectures including DGI (Velickovic et al., 2019), GRACE (Zhu et al., 2020), and BGRL (Thakoor et al., 2021) in addition to GraphMAE. We include the random initialization baseline from (Velickovic et al., 2019), which uses a randomly initialized GCN encoder to encode nodes in order to measure the quality of inductive biases in the encoder model. Baseline methods were implemented using publicly available code whenever possible. The performance of a linear classifier on raw node features and node2vec (Grover & Leskovec, 2016) embeddings is included as baselines for performance attainable from node features and graph structure alone, respectively.

For all unsupervised node classification experiments, we follow the linear classifier evaluation scheme from (Velickovic et al., 2019), where models are first trained in unsupervised fashion to learn node representations. Then, nodes are encoded, and a $\mathcal{L}_2$-normalized linear classifier is fitted on the node embeddings to measure the quality of learned node representations. We repeat training for 5 runs, and report averaged performance on each dataset. Due to the sparsity of BoW node features, we sample nonzero features with replacement for feature embedding on the forward pass through AMPNet, rather than embedding all features. This saves computations, and represents nodes with their present features as they participate in cross-attention within AMPNet. Wherever possible, we follow the specified parameters for each architecture on each dataset if reported in their respective training procedure. All training experiments are done on an NVIDIA rtx5000 GPU with 16GB of memory. To ensure that all methods fit within memory constraints, we perform subgraph sampling using GraphSAINT random walk-based sampler (Zeng et al., 2019) to avoid computing message-passing over the entire graph at once.

Table 4 summarizes the results for unsupervised node classification. We find that AMPNet performs on par with strong self-supervised approaches on benchmark datasets where nodes consist of bag-of-words features. We note that on these datasets, there is no explicit additional information to be encoded for each individual feature, and the task therefore does not take advantage of AMPNet's

expressiveness in representing individual node features. We believe that AMPNet is more well-suited for tasks where rich information can be encoded per node and node feature. AMPNet's comparable performance on standard datasets, however, suggests that it may be used in conjunction with other GNN layers in more complex architectures in order to introduce interpretability and more expressive feature encoding.

## A.4 ATTENTION CASE STUDY 3: WORD INTERACTIONS IN CITATION NETWORKS

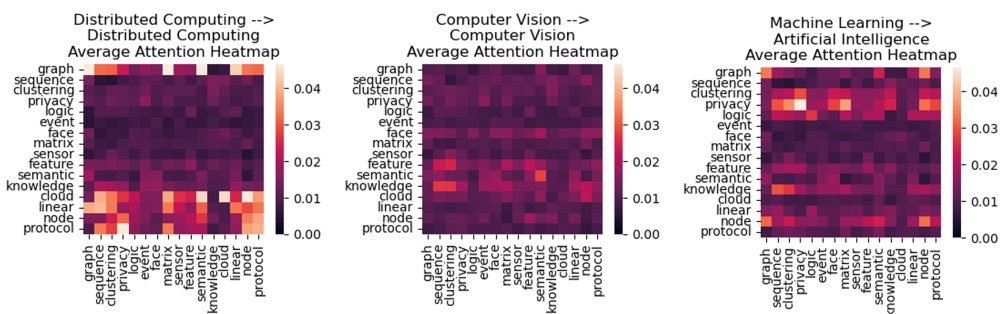

Figure 5: Averaged attention heatmap between the top 15 highly-attended words across four pairs of connected paper categories. Attention heatmaps are averaged across all edges connecting the pair of document categories. Rows represent word features of source nodes, and columns represent word features of the destination node on that edge.

Given two nodes in a citation network, where nodes are represented by embedded BoW features, AMPNet provides cross-attention matrices representing interactions between word features in neighboring nodes. To qualitatively assess the word feature interactions, we visualize four examples of cross-attention on the OGBN-arXiv dataset (Wang et al., 2020) in Figure 5. Each heatmap is calculated by averaging attention heatmaps over all edges connecting that pair of computer science document categories. Note that because cross-attention is not completely symmetric, the feature interaction on edges passing information from one category to another might not be the same if the direction of the edge is reversed.

Intuitively, the feature interaction between two categories should highlight words which appear often in both categories of documents. We see in Figure 5 that a row-wise pattern emerges in the cross attention, where destination node features (columns) nearly all pay higher attention to a select few features in the source nodes. On self-edges between papers in the Distributed Computing category, we see words such as *graph*, *cloud*, and *protocol* highlighted, which often appear when describing distributed protocols. Self-edges between Computer Vision papers highlight completely different words, with *semantic* and *feature* receiving more attention by destination node features. On cross-category edges we see more spread in which words are highly-attended to, possibly because the set of common words between inter-category documents is more disjoint compared to intra-category documents.

We note that the feature-level interactions uncovered by AMPNet depend highly on the identity of the features in the source and destination node, resulting in different highlighted features across edges when examining the same 15 words. We believe that this expressiveness in feature interaction benefits AMPNet's performance in tasks where feature interactions play important role in node-level or feature-level tasks.

## A.5 HYPERPARAMETER CONFIGURATIONS

Hyperparameter tuning was done for all architectures on the fMRI reconstruction task, Slideseq-V2 and 10X human heart masked gene expression prediction tasks through a grid search over values for learning rate, weight decay, dropout, and attention dropout where applicable. Hidden dimension was set for each model to ensure an equal number of trainable parameters across different architectures to give all models equal capacity.

Table 5: Hyperparameters used on the fMRI recording reconstruction task.

| | AMPNet | GCN | GraphSAGE | GAT | GraphMAE | GPS GT |
|---|---|---|---|---|---|---|
| Trainable parameters | 84988 | 86608 | 85520 | 86992 | 88020 | 86372 |
| Epochs | 100 | 100 | 100 | 100 | 100 | 100 |
| Learning rate | 0.003 | 0.001 | 0.001 | 0.003 | 0.001 | 0.001 |
| Dropout | 0.0 | 0.0 | 0.0 | 0.0 | N/A | 0.0 |
| Weight decay | 1e-5 | 0.0 | 0.0 | 1e-5 | 1e-5 | 1e-5 |
| Attention dropout | 0.0 | N/A | N/A | 0.0 | 0.0 | N/A |

Table 6: Hyperparameters used on the Slideseq-V2 masked gene expression prediction task.

| | AMPNet | GCN | GraphSAGE | GAT | GraphMAE |
|---|---|---|---|---|---|
| Trainable parameters | 314369 | 325680 | 317404 | 325840 | 333084 |
| Epochs | 200 | 200 | 200 | 200 | 200 |
| Learning rate | 0.003 | 0.003 | 0.01 | 0.01 | 0.01 |
| Dropout | 0.2 | 0.0 | 0.0 | 0.0 | 0.0 |
| Weight decay | 1e-5 | 1e-5 | 1e-5 | 0.0 | 0.0 |
| Attention dropout | 0.0 | N/A | N/A | 0.2 | 0.2 |

Table 7: Hyperparameters used on the 10X human heart masked gene expression prediction task.

| | AMPNet | GCN | GraphSAGE | GAT | GraphMAE |
|---|---|---|---|---|---|
| Trainable parameters | 1284793 | 1309966 | 1309798 | 1166406 | 1330668 |
| Epochs | 200 | 200 | 200 | 200 | 200 |
| Learning rate | 0.01 | 0.001 | 0.003 | 0.001 | 0.01 |
| Dropout | 0.0 | 0.2 | 0.0 | 0.0 | 0.2 |
| Weight decay | 0.0 | 1e-5 | 0.0 | 1e-5 | 0.0 |
| Attention dropout | 0.2 | N/A | N/A | 0.0 | 0.2 |