# OpenReview forum: "AMPNet: Attention as Message Passing for Graph Neural Networks"
_ICLR.cc/2024/Conference — ICLR 2024 Conference Withdrawn Submission_

### Official Review · Reviewer_Kcsy · 2023-10-27

**Soundness:** 2 fair
**Presentation:** 2 fair
**Contribution:** 1 poor
**Rating:** 3
**Confidence:** 5

**Summary:**

This paper proposes a new GNN layer called AMPNet that encodes individual features per node and models feature-level interactions through cross-node attention during message-passing steps.  The effectiveness of AMPNet is evaluated on fMRI recording, spatial genomics datasets, obgn-arxiv etc, and the auhtors also present case studies to try to provide interpretability.

**Strengths:**

1. This paper proposes a new GNN layer called AMPNet, which is claimed to model intricate details about individual node features.

2. AMPNet achieves considerable improvements on fMRI brain activity recordings and spatial genomic data.

**Weaknesses:**

1. The motivation for AMPNet is not clear. The authors claim that conventional GNNs overlook intricate details about individual node features by representing each node with a singular feature vector. However, they do not provide any concrete evidence to support this claim.

2. The work is really incremental. Basically, AMPNet just transforms node feature $x_v \in R^F$ to $H_v \in R^{F \times D}$, then follows the standard multi-head attention.

3. The experiments results are not convincing, since the baselines are quite classical (gcn, gat, graphSAGE, etc ). SOTAs are prefered on each dataset.

**Questions:**

1. A case study or analysis on why existing GNN cannot model intricate details about individual node features.

2. Comparisons on SOTA approaches on the experimental datasets.

3. The ogbn-arxiv is for node level tasks, why choose this dataset for link prediction?

4. The related work section mainly discuss interpretability related literatures, then is the interpretability issue the focus of this paper? If so, the authors should clearly state that in the abstract.

---

### Official Review · Reviewer_vyBv · 2023-10-31

**Soundness:** 2 fair
**Presentation:** 3 good
**Contribution:** 1 poor
**Rating:** 3
**Confidence:** 4

**Summary:**

This paper introduces an attention mechanism between the features of nodes-to-aggregate in GNNs.
The basic idea is to adopt the well-known QKV learnable mechanism from Transformers, but applied to
understand the relationships between the features of nodes-to-aggregate. The authors motivate
the paper as a means of explaining how message passing uncovers certain relationships between
neighboring nodes.

The method is compared with some baselines, including Transformers and the authors highlight a couple
of case studies (fMRI-brain and Genomic-data). Results in link-prediction are included as well.

Summarizing this is a basic attention method for GNNs and the paper discloses its utility in several domains.

**Strengths:**

* The method shows the applicability of GNNs to several domains (neuroscience, genomics).
* The contribution is interesting but very basic. Attention mechanisms are well-known.

**Weaknesses:**

* Experimental results are very terse: basic baselines.

**Questions:**

How does this mechanism avoid any GNN critical point such as over-smoothing?
How does this method compare with stronger GNNs (e.g. GIN). More baselines are needed, please.

---

### Official Review · Reviewer_Jr5A · 2023-11-01

**Soundness:** 2 fair
**Presentation:** 3 good
**Contribution:** 3 good
**Rating:** 5
**Confidence:** 4

**Summary:**

The paper presents AMPNet, a novel GNN architecture that introduces an attention-based message-passing layer, which allows AMPNet to model feature-level interactions through cross-node attention. This approach not only enables the model to encode individual features per node but it also help the interpretability of GNNs.

**Strengths:**

1. The paper is overall well-written and easy to follow. The authors also provide enough details about the methodology.
2. The model's ability to provide interpretable feature-level interactions is a notable advantage, particularly in healthcare related domains, which require interpretable models.
3. The paper demonstrates that AMPNet outperforms existing models in fMRI reconstruction and gene expression prediction.

**Weaknesses:**

1. Recently, there have been several graph-learning models to capture cross-node feature dependencies using mlp-mixer (e.g., [1, 2, 3]). There is a lack of discussion on these methods in the paper. It would be better if the author could also include some of these methods as the baselines since they aim to address almost a similar challenge as this paper.
2. Current experiments are limited to fMRI and gene networks, which makes it unclear that how this approach performs in other domains. Is  AMPNet only good for fMRI and gene analysis? If so, it would be better to include state-of-the-art fMRI encoding (specifically [2], which tries to address cross-node feature dependencies as well.) and gene expression prediction methods. If it is not limited to these domains, it would be better to add more real-world datasets from OGB [4].
3. It would be better if the authors could perform an ablation study and parameter sensitivity to discuss some experimental and model design choices.


[1] A Generalization of ViT/MLP-Mixer to Graphs. ICML 2023.

[2] ADMIRE++: Explainable Anomaly Detection in the Human Brain via Inductive Learning on Temporal Multiplex Networks. ICML IMLH 2023.

[3] Do We Really Need Complicated Model Architectures For Temporal Networks? ICLR 2023.

[4] Open Graph Benchmark: Datasets for Machine Learning on Graphs. NeurIPS 2020.

**Questions:**

Please see Weaknesses.

---

### Official Review · Reviewer_QxkE · 2023-11-06

**Soundness:** 2 fair
**Presentation:** 2 fair
**Contribution:** 2 fair
**Rating:** 3
**Confidence:** 4

**Summary:**

The paper introduces a new architecture called AMPNet (Attention-based Message-Passing Network) that enhances the representational power of Graph Neural Networks. It achieves this by encoding individual node features and utilizing cross-node attention to facilitate feature-level interactions. The performance of AMPNet has been benchmarked on complex biological datasets, including fMRI brain activity and genomic data. The results demonstrate a remarkable improvement of 20% in fMRI reconstruction accuracy compared to existing models. When incorporating positional embeddings, the improvement is further increased by 8%. The paper also presents case studies in biological domains that highlight the practical usefulness of AMPNet in graph-structured data that is rich in feature-level detail.

**Strengths:**

1. Clarity of Writing: Despite the missing appendix, the paper is well-written with a clear flow, making it accessible and easy to follow.
2. Addressing the Need for Explainability: The paper tries to address a highly relevant and challenging issue in graph learning – the need for explainability in GNN methods.
3. Exploring Self-supervised Reconstruction: The paper's exploration of self-supervised reconstruction tasks positions it at the forefront of innovative research directions in graph learning. Self-supervised learning is indeed a potent strategy for pretraining models, and its application within graph neural networks could pave the way for more robust and generalizable GNN architectures.

**Weaknesses:**

1. Missing Appendix: The paper is currently missing its appendix, which is crucial for providing additional methodological details. This omission is significant as it leaves questions unanswered regarding the specifics of the loss function employed and the particularities of how different reconstruction tasks are defined across various datasets. The absence of this detailed exposition can severely hinder the reproducibility and thorough understanding of the proposed method.
2. Limited Technical Innovation: The technical novelty of the proposed method is called into question due to its reliance on a conventional attention mechanism for neighbor node embedding fusion. The expansion of node features from one dimension to two does not show enough technical contribution.
3. Insufficient Demonstration of Interpretability: Despite claims of improved interpretability, the paper does not convincingly showcase this quality through its case study results (Also see Question 2)

**Questions:**

1. The submission misses the Appendix part. Therefore, there is a lack of a lot of experimental details.
2. The method part emphasizes the model can "provide interpretable coefficients for feature-level interactions between nodes," but in Figure 3, the shown attention is based on node-wise attention. Can you explain how to convert the feature-level attention to node-level attention?
3. What is the time series represented in the fMRI dataset?